# Informing climate-health adaptation options through mapping the needs and potential for integrated climate-driven early warning forecasting systems in South Asia—A scoping review

**Festus A. Asaaga** [ORCID]**\*, Emmanuel S. Tomude\*, Nathan J. Rickards, Richard Hassall, Sunita Sarkar, Bethan V. Purse**

UK Centre for Ecology & Hydrology, Wallingford, United Kingdom

\* fesasa@ceh.ac.uk (FAA); emmanuel.stomude@gmail.com (EST)

## Abstract

### Background

Climate change is widely recognised to threaten human health, wellbeing and livelihoods, including through its effects on the emergence, spread and burdens of climate–and water-sensitive infectious diseases. However, the scale and mechanisms of the impacts are uncertain and it is unclear whether existing forecasting capacities will foster successful local-level adaptation planning, particularly in climate vulnerable regions in developing countries. The purpose of this scoping review was to characterise and map priority climate- and water-sensitive diseases, map existing forecasting and surveillance systems in climate and health sectors and scope out the needs and potential to develop integrated climate-driven early warning forecasting systems for long-term adaptation planning and interventions in the south Asia region.

### Methods

We searched Web of Science Core Collection, Scopus and PubMed using title, abstract and keywords only for papers focussing on climate-and water-sensitive diseases and explicit mention of either forecasting or surveillance systems in south Asia. We conducted further internet search of relevant national climate adaptation plans and health policies affecting disease management. We identified 187 studies reporting on climate-sensitive diseases and information systems in the south Asia context published between 1992 and 2024.

### Results

We found very few robust, evidenced-based forecasting systems for climate- and water-sensitive infectious diseases, which suggests limited operationalisation of decision-support tools that could inform actions to reduce disease burdens in the region. Many of the information systems platforms identified focussed on climate-sensitive vector-borne disease

**Data Availability Statement:** All relevant data are within the manuscript and its Supporting information files.

**Funding:** The research was funded by NERC as part of National Capability. The funders had no role in study design, data collection and analysis, decision to publish, or preparation of the manuscript.

**Competing interests:** The authors have declared that no competing interests exist.

systems, with limited tools for water-sensitive diseases. This reveals an opportunity to develop tools for these neglected disease groups. Of the 34 operational platforms identified across the focal countries, only 13 (representing 38.2%) are freely available online and all were developed and implemented by the human health sector. Tools are needed for other south Asian countries (Afghanistan, Sri Lanka, Bhutan) where the risks of infectious diseases are predicted to increase substantially due to climate change, drought and shifts in human demography and use of ecosystems.

## Conclusion

Altogether, the findings highlight clear opportunities to invest in the co-development and implementation of contextually relevant climate-driven early warning tools and research priorities for disease control and adaptation planning.

## Introduction

Amidst the current and projected impacts of global climate change and longstanding water securitisation concerns, there is widespread consensus that context-specific and targeted adaptation interventions are required at all scales to alleviate risks to human health, wellbeing and livelihoods [1–4]. Recent United Nations Framework Convention on Climate Change (UNFCCC) Conference of Parties (COP26) resolutions and attendant national commitments on ecosystem restoration, add further impetus for enhancing resilience and local adaptive capacity to different climate-related risks. These risks may include alteration of ecosystems, disruption of food production and water, and consequences for livelihoods, health and wellbeing within national adaptation planning frameworks [5]. This is particularly important for low-and middle-income countries (LMICs) characterised by limited capacity to adapt to climate-related hazards (e.g., diseases and food insecurity). Available statistics indicate that about 1.2 billion Indians are directly dependent on natural ecosystems for their livelihoods and wellbeing, and disproportionately affected by zoonotic, vector-and water-borne infectious diseases [6–9].

Burgeoning evidence suggests that climate change is already driving the emergence and spread of a number of climate-related diseases [10], including malaria and meningitis [11–13]. The increasing availability of climate and land surface models and data at different spatio-temporal scales and its combination with health and socio-economic data within epidemiological models offers the opportunity to advance the understanding of links between climate change and health outcomes and to generate forecasts and risk information to inform preparedness [14–16]. However, it is critical to understand how climate-related risks to human health are prioritised, against the backdrop of wider climate-related stressors, and leverage scarce resources and information across environmental and health sectors to enhance preparedness, particularly in highly vulnerable regions in the south Asian and sub-Saharan African regions.

South Asia (comprising Afghanistan, Bangladesh, Bhutan, India, Maldives, Nepal, Pakistan, and Sri Lanka) is home to a quarter of the world's population and plagued with a considerable proportion of the global burden of infectious disease [8,17–20]. There is evidence of emerging and growing threats such as Zika, dengue, Middle East respiratory syndrome coronavirus (MERS-CoV), avian influenza and more recently coronavirus disease-2019 [21]. These emerging diseases risks add to the already significant disease burden, and their impacts may

accelerate in the face of climate change and other contextual factors such as land use intensification/change, inadequate surveillance, uneven health system capacity and deficiencies in water and sanitation [22]. This raises important questions related to the priority climate-and-water sensitive diseases across the region such as what hydro-climatic, health and agriculture information systems are available to stakeholders in the region to potentially inform their decision-making? Who are the key actors and knowledge holders, and what are the forecasting (disease and hydroclimate) capabilities and the opportunities to better link these across sectors? How and to what extent can development of integrated climate-driven early warning systems inform climate and disease adaptation planning and interventions in the region? The answers to these questions provide critical policy relevant insights on the targeting and prioritisation of interventions and development of integrated forecasting capabilities to bolster the adaptive capacity of affected or at-risk groups and communities in the sub-region [20,23–27].

Considerable literature has examined the effects of climate change upon the incidence or impacts of specific diseases in different socio-ecological settings [28], and climate- or hydrology-driven health and agricultural information and forecasting capabilities, mostly in the global North [29]. Yet, to date, except for a technical report on a global scan of software tools for climate-sensitive infectious diseases modelling reported between 2011 and 2021 [30], there has not been any systematic assessment of the priority climate-and-water sensitive diseases, cross-sectoral information and forecasting and surveillance capabilities, and stakeholder needs in the South Asia context. We have limited context-specific understanding of the links between hydrological systems, environmental processes, and ecosystem changes that shape disease dynamics in terms of emergence, establishment and spread, especially in the climate- vulnerable regions in South Asia [31,32]. The Indian sub-continent, for instance, is already experiencing variability in climate and the impacts of climate change, including water stress, heat waves and droughts, severe storms and flooding and associated negative consequences on health and livelihoods [9,33]. This further amplifies the need for targeted and contextually relevant interventions to bolster local-level adaptation to projected climate impacts and associated risks on human health and livelihoods [34–36].

Some successful climate-driven early warning systems have been developed for climate-and-water sensitive diseases [37–40], often delivering to sub-national, national and regional level decision-makers in disease management and focussed on a narrow range of climate-related risks. Some authors have also observed that these systems may be poorly contextualised and not penetrate to community level decision makers [30,41,42]. This highlights the wider disconnect between models, policy and interventions among the climate and epidemiological research communities [43,44]. There is, therefore, potential to integrate a broader range of data and expertise across sectors to improve the forecasting ability of early warning systems and to develop such systems for a larger range of diseases [30]. However, there also remains a disconnect between science and decision-making policy in public health and agriculture, particularly for zoonotic diseases [45]. Recent efforts to bridge these gaps for zoonotic diseases have included participatory modelling, co-production and integration of community perspectives for improved contextualisation of models [45–50]. Here, therefore, we map the priority climate-and water sensitive diseases, existing cross-sectoral forecasting capabilities, and explore the potential for developing integrated climate-driven early warning forecasting systems to support context-specific adaptation interventions for climate-and water sensitive risks to health in south Asia.

## Methods

We conducted a scoping review of published literature in international refereed journals to characterise the research evidence, gaps and potential for building integrated climate-driven

early warning forecasting systems following the Preferred Reporting Items for Systematic Review and Meta-Analysis (PRISMA) guidelines [51].

## Search strategy—Identifying relevant studies and national policies

We developed a search protocol which generated a list of possible search strings (related to climate-and water sensitive diseases in the south Asia context) used to locate relevant published literature. A pilot search was conducted in Web of Science Core Collection using a defined search-string, based on which iterations of search terms were developed (see S1 File). This was done using the authors' expert knowledge and reference to literature. We adopted a broad search approach to allow for a comprehensive focus and retrieval of papers across different disciplines. No timeframe restriction was applied.

For the actual systematic search, the online bibliographic databases of PubMed, Web of Science Core Collections and Scopus were used to identify relevant peer-reviewed articles, given their wide scope of scientific publications and multidisciplinary contents [52]. In addition, the reference lists of randomly selected articles were manually searched using a "snowball" technique to identify any further literature that may have been missed in the first round of searches. This was repeated until saturation of the search had been reached. The key terms applied in the search are summarised in the accompanying excel scoring sheet (see S1 File). The search results were imported into Sciwheel reference management software (https://sciwheel.com) and duplicate articles were automatically removed.

Following the literature search, we conducted an online search of relevant national climate adaptation plans, health policies and hydroclimate and health surveillance systems of the focal countries. The keywords "climate policy", "health policy or legislation", "climate change adaptation policy", "hydroclimate information systems", "health surveillance systems" "climate-sensitive diseases" and "country" were used to search documents about policies and legislations of relevance to the subject-matter. The rationale for the policy mapping exercise was to establish the extent of national policy focus on climate-and-water sensitive diseases, and the opportunities for operationalising climate-driven early warning forecasting systems. This also afforded contextual insights into respective national government priorities and investment focus concerning tackling climate-and-water sensitive human disease risks as part of national climate-health adaptation planning. National policies in this context comprise the set of guidelines and legislative enactments adopted by the focal countries to support climate change adaptation, health and development planning. National adaptation policy thus refers to a formal national policy for identifying medium- and long-term adaptation needs and developing and implementing strategies and programmes to address them [53]. Source documents were collated from relevant government ministry and departmental websites for critical analysis and assessment of gaps. Both the peer-reviewed literature and policy searches were conducted between 2$^{nd}$ January 2022 and 30th April 2022. A further peer-reviewed literature search was carried out (using the same keyword search terms) between 23$^{rd}$ January 2024 and 1$^{st}$ February 2024 based on which the corpus of relevant literature was updated.

**Eligibility criteria.** After the systematic search, different inclusion and exclusion criteria were used to select peer-reviewed articles and policy documents for the review (Fig 2). Only electronically available, full text articles and policies written in English were considered. The identified articles were screened for eligibility in two phases. In the first phase, studies that did not report on forecasting systems linked to climate and/or water sensitive diseases and south Asia were excluded. If the article title and abstract information was insufficient relative to the inclusion/exclusion criteria, we retained the article in the Sciwheel database for further full text review. At the full text review stage, we further excluded articles that were inaccessible. In total

1,091 citations and 40 national climate and health policies and reports (latter are listed in Table 1) were imported to the Sciwheel database for the review.

**Data extraction and thematic analysis.** Following the selection of relevant studies, a thematic and content analysis approach was used to extract and organise relevant information based on a deductive coding protocol created in online surveys (https://www.onlinesurveys.ac.uk/). We sourced the full text of 261 relevant studies (published between 1992 and 2024) across 107 refereed journals. Using a thematic and content analysis approach [54], we extracted information across three central domains: (1) study background information (title, authorship, country/region and journal), (2) study design and methodology (sampling approach, population/sample, methods, scale), and (3) study findings and conclusions (disease/pathogen investigated, transmission pathways, climate & land use drivers, health surveillance/ information systems, disease risk factors).

Concerning the policy review, we conducted a detailed textual analysis on the sourced policy documents of relevance to climate and health linkages. We developed an initial coding framework (based on expert knowledge) comprising four thematic areas of focus for information extraction: (1) background information about the policy document (date of entry into force, aims and objectives relating to climate-health adaptation), (2) sectoral representation, affected community and scale of implementation, (3) explicit focus on priority climate-and-water sensitive diseases, and (4) gaps in policy coverage. We developed a policy-scoring matrix to assess if policy statements make mention of climate-and water sensitive diseases and rationale of focus. For instance, if a policy is scored 'yes' for mention any climate-sensitive disease, then another column investigates the specific interest in the said disease. Where appropriate, we analysed the results of different types of diseases, divided by their causal agent (bacteria, viruses, parasites, protozoans), disease status (emerging, epidemic or endemic in a given country or region) or by their transmission pathway (see Table 1 legend), drawing information from WHO, the US CDC and national CDCs [10,55]. We use the term driver to mean the underlying mix of antecedent epidemiological conditions that are necessary for a pathogen to emerge in susceptible population (for emerging, or epidemic diseases [56]) or for a pathogen to increase in impact in a population (for re-emerging or endemic diseases. We define climate drivers as climate-related hydrometeorological hazards that impact on the focal disease system, by affecting demographic rates of pathogens, vectors or hosts directly or indirectly via the resources that they utilise [57]. We define non-climatic drivers, as those that are unrelated to hydrometeorological hazards, such as land use and socio-economic factors, that again can act directly or indirectly on the disease system.

## Results and discussion

The search strategy retrieved 1,091 (Web of Science = 998; PubMed = 59; Scopus = 34) peer-reviewed articles from three databases (Fig 1). After duplicates removal, the titles and abstracts of 862 publications were screened. From the title and abstract screening, only 261 studies were included for full-text screening and 187 articles met the inclusion criteria.

### The literature landscape

As illustrated in Fig 1, we identified 862 citations after removing duplicates, and 274 citations were selected for the abstract and full-text screening. After concluding the two-phase screening process, 187 articles published between 1992 and 2024 (120 studies between 2015 and 2024) were retained for the review, most of which were empirical studies (Figs 1 and 2A, n = 144). Of the 187 studies, 175 were country-specific and 12 covered the south Asia region (multi-country; Fig 2B). More than half of the selected studies (54.5% or n = 102 studies) came from study

**Table 1. Key national climate and health policies in the reviewed countries.**

| Country | Policy | Objective | Priority disease focus and frequency of explicitly mentions of individual disease in respective documents (in parentheses) |
|---|---|---|---|
| Afghanistan | Climate Change and Governance in Afghanistan 2015 | To strengthen the need for climate change adaptation | Infectious diseases (1 mention), pest and diseases (3 mentions) |
| | Second National Communication under the United Nations Framework Convention on Climate Change (UNFCCC) 2017 | Aim to strengthen Afghanistan's efforts in climate change mitigation with reference to the vision of UNFCCC. | Malaria[m] (5 mentions); leishmaniasis[s] and cholera[w] (2 mentions); taeniasis, ascariasis and diarrhoea[w] (1 mention respectively) |
| | National Health Strategy 2016–2020- Sustaining Progress and Building for Tomorrow and Beyond | to prevent ill health and achieve significant reductions in mortality in line with the national targets and sustainable development goals and to reduce impoverishment due to catastrophic health expenditure | Malaria (4 mentions); leishmaniasis (1 mention) |
| Bangladesh | Bangladesh Climate Change and Sustainable Development (2000–2050) | Identifies various climatic factors, possible climate change scenarios and adaptation possibilities. | Diarrhoea (10 mentions); dengue (9 mentions); malaria (8 mentions); cholera (7 mentions); Lymphatic filariasis[m] and helminthiasis (3 mentions) |
| | Nationally Determined Contribution 2021 | Aims to further strengthen mitigation actions that Bangladesh may take to tackle its growing emissions and to play its role in global efforts to limit temperature rise to 2 degrees or preferably 1.5 degrees Celsius above pre-industrial levels. | Not mentioned |
| | Bangladesh Climate Change Strategy and Action Plan 2009 | A ten-year programme to build the capacity and resilience of the country to meet the challenge of climate change over the next 20-25years. | Not mentioned |
| | Climate Change Profile 2018 | To help integrate climate actions into development activities. | Pest and disease (2 mentions); cholera and malaria (1 mention) |
| | Economics of adaptation to climate change 2010 | To help decision makers in Bangladesh to better understand and assess the risks posed by climate change and to better design strategies to adapt to climate change. | Not mentioned |
| | Evaluation of Adaptation Policies in GBM Delta of Bangladesh 2018 | Aims at creating an enabling environment for sustainable growth of agriculture for reducing poverty and ensuring food security through increased crop production and employment opportunity. | Water-borne diseases (1 mention) |
| Bhutan | Third National Communication to the UNFCCC 2020 | The Third National Communication from Bhutan to the UNFCCC elaborates the actions taken, which are required in addition to emission mitigation and address adverse impacts of climate change in Bhutan | Dengue (20 mentions); malaria (18 mentions); chikungunya[m] (2 mentions) |
| | Strategizing Climate Change for Bhutan 2009 | Consider issues such as sustainable land management, climate change, disaster management, energy sustainability, globalization and infrastructure development in order to guide sectors in the formation of their environment chapters for the 10th Five Year Plan. | Malaria (3 mentions); dengue (1 mention) |
| | Climate Change Policy of the Kingdom of Bhutan 2020 | To provide strategic guidance to ensure that Bhutan remains carbon neutral and protect the wellbeing of the people of Bhutan by adapting to climate change in an efficient and effective manner. | Not mentioned |
| | Second Nationally Determined Contribution 2021 | Strengthening forest management practices, climate-smart primary production, integrated land use planning, and improved rural livelihoods. | Not mentioned |
| | Bhutan National Adaptation Programme of Action (undated) | The NAPA process aims to have an uncomplicated and efficient approach to addressing the most urgent and immediate needs of the vulnerable communities. | Vector-borne disease in wildlife, malaria (2 mentions), dengue (1 mention), water-borne disease (unspecified), pest and diseases (unspecified) |
| | From global ambition to country action; Bhutan steps towards low—carbon climate—resilient development 2018 | Aims to enhance national, local and community capacity to prepare for and respond to climate induced hazards | Mosquito diseases (no specific disease mentioned) |

*(Continued)*

**Table 1.** (Continued)

| Country | Policy | Objective | Priority disease focus and frequency of explicitly mentions of individual disease in respective documents (in parentheses) |
|---|---|---|---|
| India | Climate Change and Human Health—A report of the DST's National Knowledge Network Programme on Climate Change and Human Health 2016 | Aims at developing strategic knowledge on key climate change issues that include socio-economic sectors impacted by climate change. | Malaria (183 mentions); dengue (4 mentions); cholera (1 mention) |
| | National Mission on Strategic Knowledge for Climate Change 2010 | Formation of knowledge networks among the existing knowledge institutions engaged in research and development relating to climate science and facilitate data sharing and exchange through a suitable policy framework and institutional support | Pest disease (not specific)-7 mentions |
| | National Action Plan on Climate Change (undated) | Aims to promote overall economic development and improve the socioeconomic conditions of the resource poor and disadvantaged sections inhabiting the programme areas. | Malaria (2 mentions); visceral leishmaniasis[s], Japanese encephalitis[m], Lymphatic filariasis, and dengue (1 mention respectively); pest and disease (unspecified) |
| | National Health Policy of India 2015 | Addresses the urgent need to improve the performance of health systems | Malaria (6 mentions); visceral leishmaniasis[s] (3 mentions), Lymphatic filariasis and Japanese encephalitis (1 mention respectively) |
| | Electronic health record (EHR) standards for India 2016 | The primary aim of interoperability standards is to always ensure syntactic (structural) and semantic (inherent meaning) interoperability of data amongst systems. | Communicable diseases (not specific)-1 mention |
| Nepal | National Strategy for Sustainable Development (NSSD) 2001 | To reduce poverty through the implementation of programmes in the identified priority sectors such as agriculture; water resources; social sector; industry, tourism and international trade; and physical infrastructure. | Disease and pest infestation (not specific) |
| | Country Cooperation Strategy 2018 | Aims at immunizing all children through a strong community engagement. | Malaria (9 mentions); visceral leishmaniasis and cholera (2 mentions respectively); Lymphatic filariasis (1) |
| | Nepal Health Sector Support Programme 2018 | To review the existing policies in the health sector to inform policy development and revisions in the federal context | Zoonotic diseases (not specific) |
| | Nepal Malaria Strategic Plan 2014–2025 | To achieve zero deaths due to malaria by 2015 and sustaining, reducing the incidence of indigenous malaria cases by 90% | Malaria (943 mentions); leptospirosis (9 mentions); dengue (8 mentions); Lymphatic filariasis (2 mentions); chikungunya and, Japanese encephalitis (1 mention respectively) |
| | National Climate Change Policy, 2076 (2019) | Building a climate resilient society through climate change and mitigation and adaptation | Forest and pest diseases (not specific)-1 mention |
| Pakistan | National Health Policy 2009 | The goal of the national health policy is to remove barriers to access to affordable, essential health service for every Pakistani | Dengue (6 mentions); malaria (5 mentions), avian influenza (1 mention) |
| | National Health Vision Pakistan 2016 (2016–2025) | To improve the health of all Pakistanis, particularly women and children, through universal access to affordable quality essential health services, and delivered through resilient and responsive health system, ready to attain Sustainable Development Goals and fulfil its other global health responsibilities. | Malaria (7 mentions) and tuberculosis (2 mentions) |
| | Updated Nationally Determined Contribution 2021 | Aims at achieving reduced poverty and ensuring a stable economy. | Dengue (1 mention) |
| | National Climate Change Policy 2012 | To ensure that climate change is mainstreamed in the economically and socially vulnerable sectors of the economy and to steer Pakistan towards climate resilient development | Pest and disease (2 mentions); malaria and dengue (1 mention respectively) |
| | Framework for Implementation of Climate Change Policy (2014–2030) | Building capacities of national institutions, provincial forest departments and other stakeholders for effective development and implementation of innovative mechanisms aiming at avoiding deforestation and enhancing forest carbon stocks. | Malaria (3 mentions); dengue (2 mentions) |

*(Continued)*

**Table 1.** (Continued)

| Country | Policy | Objective | Priority disease focus and frequency of explicitly mentions of individual disease in respective documents (in parentheses) |
|---|---|---|---|
| Sri Lanka | Policy Repository of Ministry of Health Sri Lanka 2016 | Aims at increasing life expectancy by reducing preventable deaths due to both communicable and non-communicable disease, improve the quality of life by reducing preventable diseases, health problems and disability | Rabies (215 mentions); malaria (7 mentions) |
| | National Health Policy 2016 (2016–2025) | To contribute to social and economic development of Sri Lanka by achieving the highest attainable health status. | Malaria (6 mentions); dengue (5 mentions); water-borne diseases and Lymphatic filariasis (3 mentions respectively), rabies (2 mentions) |
| | Updated Nationally Determined Contributions 2021 | To facilitate sustainable development in each sector in a way that supports continued economic growth and high human development while protecting the natural resource base on which many of these livelihoods are dependent. | Malaria (3 mentions); pest and diseases (3 mentions) |
| | National Strategic Framework for Development of Health Services 2016 (2016–2025) | To safeguard the status of Health of all citizens of Sri Lanka | Tuberculosis (10 mentions); dengue (6 mentions); malaria (5 mentions); leishmaniasis (4 mentions); leptospirosis$^w$ (2 mentions); Lymphatic filariasis (1 mention) |
| | National Climate Change Adaptation Strategy for Sri Lanka 2016 (2016–2025) | To increase the resilience of economic sectors and natural systems against the emerging and projected impacts of climate change by adopting appropriate coping strategies and system improvements | Vector-borne diseases (11 mentions); specific disease mentioned—dengue (1 mention) |
| | National Health Strategic Master Plan 2016 (2016–2025) | To safeguard the status of Health of all citizens of Sri Lanka | Food or waterborne (6 mentions), vector borne disease (1 mention) |
| | Health System Review 2021 | To provide a tool for the dissemination of information on health systems and the exchange of experiences between policymakers and analysts in different countries implementing reform strategies; and to assist other researchers in more in-depth comparative health policy analysis. | Dengue (14 mentions); tuberculosis (10 mentions); leptospirosis (6 mentions); pandemic influenza (7 mentions) |
| | Climate Change Policy (undated) | Deal with the numerous facets of climate change issues | Dengue (1 mention) |
| Maldives | Policy brief for Maldives | Evaluate south Asia's Current Community Health Worker Policies and System Support and their Readiness for Community Health Workers' | Vector borne disease, infectious disease, neglected tropical diseases |
| | Second National Communication of Maldives to the United Nations Framework Convention on Climate Change | builds on the activities undertaken during the FNC and provides the developments in efforts made to address climate change in the Maldives | Dengue (19 mentions); Chikungunya (11 mentions); Scrub-typhus$^{mi}$ (5 mentions); Diarrhoeal (4 mentions); malaria (3 mentions); Lymphatic filariasis (2 mentions) |

*Superscript m = mosquito-borne disease, mi = mite-borne disease; s = sand-fly-borne disease; w = water-borne disease.

PRISMA 2020 flow diagram for updated systematic reviews which included searches of databases, registers and other sources

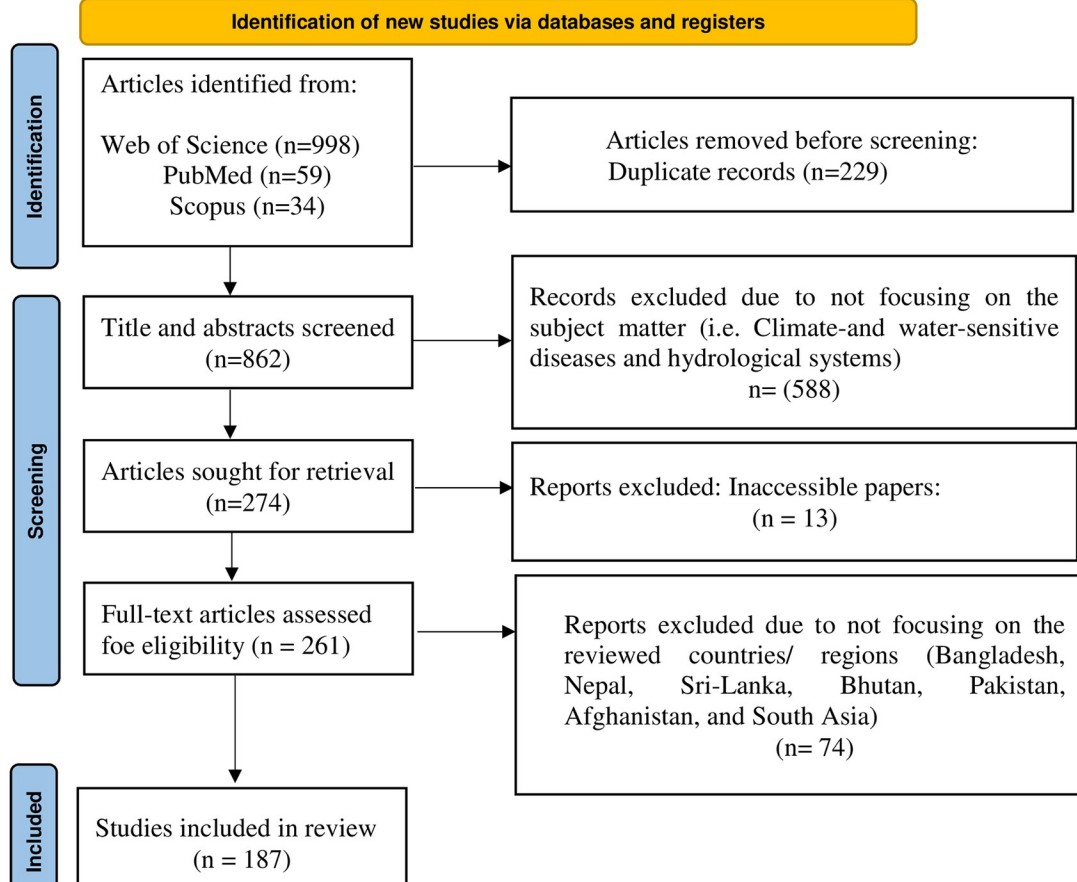

**Fig 1. PRISMA 2020 flow diagram.** Adapted from the Preferred Reporting Items for Systematic Review and Meta-Analysis (PRISMA) protocol by Page et al. [51].

locations in India, followed by Bangladesh (20.9% or n = 39 studies), Sri Lanka (9.1% or n = 17 studies), Nepal (6.9% or n = 13 studies), and Pakistan (3.7% or n = 7 studies), as the top-most countries of research focus (Fig 2B). Bhutan (n = 2 studies) and Afghanistan (n = 1 study) represented the countries of the least research focus (for details of the papers, see Fig 2 and S1 Table (country distribution of included studies) in the Supplementary Information). The frequency of publications over time indicates that the number of climate-health studies from the south Asia region remained very low until 2012, followed by a steady increase, reaching maximum numbers in 2020 coincident with the publication of the Lancet 2020 seminal report dubbed the Lancet Countdown on health and climate change: responding to converging crises (Fig 2A).

Concerning study design, we found a strong focus on studies adopting quantitative approaches (e.g. modelling) with 64.2% (n = 120), whereas 22.2% of studies (n = 43) employed qualitative-based (e.g. key-informant interviews, textual analysis), and 13.6% of studies (n = 24) mixed methods approaches (Fig 2C). This can be explained by the fact that most of the studies sought to quantify the degree of sensitivity of emerging infectious diseases to climate-related impacts. The flipside of this 'quantitative' focus, however, is the scarcity of

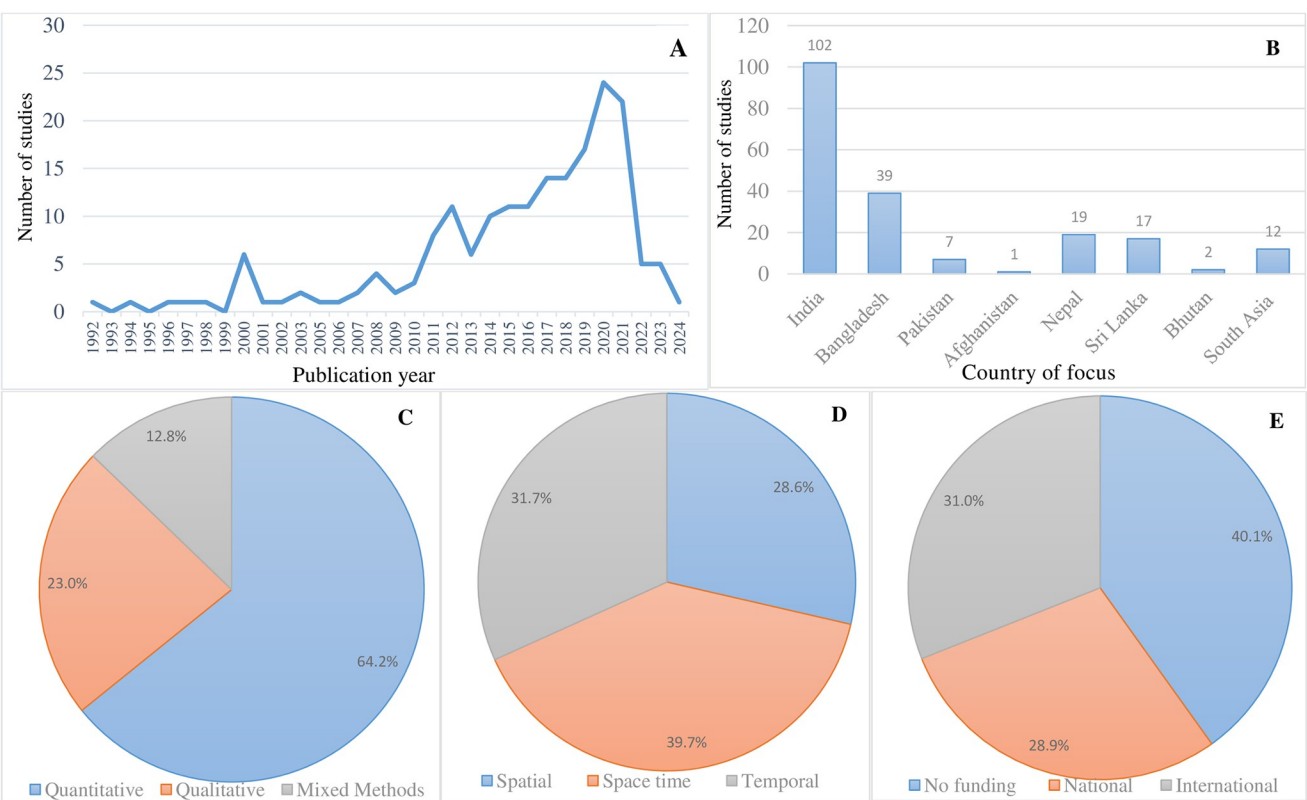

**Fig 2. Descriptive summary of included articles.** (A) For relevant abstracts, trends in publication over time indicate a continued increase in the volume of literature on climate and health in south Asia region. Literature published between 1st January 1992, and 31st January 2024 were included. (B) Study countries–each represented country is mentioned in at least one study. Pie charts show the number of publications by study design (C), model type for the modelling-based papers (D) and funding source reported (E).

interdisciplinary studies on the theme of climate-health, and therefore the understanding of complex socio-ecological and contextual drivers of vulnerability and adaptation to emerging climate-and water-sensitive diseases in the region. Of the 120 quantitative papers, slightly over half (52.5% or n = 63 studies) were modelling studies. Further disaggregation by modelling design indicates that 39.7% (n = 25 studies) of these 63 papers were space-time and temporal, with 28.6% (n = 18 studies) spatially focussed (Fig 2D). In terms of model type, nearly three-quarters of the modelling studies were correlative (71.4% or n = 45 studies) and 25.4% process-based (n = 16 studies), which is as expected since process-based models can only be parameterised if there is a sufficient information on the study system.

The reviewed papers were published in a diverse corpus of refereed journals, with 13 studies published in Malaria journal, 7 studies in Tropical Medicine & International Health, and 5 studies in each of Acta Tropica and Indian Journal of Medical Research. Further descriptive analysis of the citation bibliometrics indicate that 84 studies (representing 44.9% of the reviewed papers) involved international collaborations, meaning that one or more of the co-authors had their institutional affiliations outside of the reviewed countries. Of the 112 studies with funding disclosure statements, 58 studies (51.7%) were funded by external institutions based in Global North countries (US, Germany and UK) and supra-national institutions (World Health Organisation, United Nations Development Programme and World Bank) (Fig 2E). Most of the externally funded studies received support specifically from the Bill & Melinda

Gates Foundation and joint WHO/TDR/UNDP/World Bank scheme. This perhaps suggests a future window of opportunity for increased funding support for more international collaborative research in the development of climate-driven early warning decision tools to support ongoing adaptation and health interventions in the region. Most of the funded studies were from Bangladesh and India (n = 30 studies apiece), Sri Lanka (n = 9 studies) and Nepal (n = 8 students). Pakistan, Afghanistan and Bhutan tended to lag in terms of international research funding in the climate-health theme, which perhaps is suggestive of priority contexts for future international funding investment. From a country perspective, we found 36.7% of the funded studies from India (n = 11 studies) were supported by the Indian Council for Medical Research (ICMR), the country's flagship organisation for the formulation, coordination and promotion of biomedical research. A further cross-tabulation analysis of the authorship affiliation and publication year reveal a steady rise in the number of papers involving international collaborations from 2014 onwards (76 studies in total). The emphasis of climate change impacts on health in the IPCC 5th Assessment Report and 2015 Lancet Commission Report on Climate and Health may have contributed to the growing research interest on the climate-health theme in the region.

## Policy prioritisation of climate-and water-sensitive diseases

To understand the degree of representation or mention of climate-and-water sensitive diseases as a function of their prioritisation in policy, we reviewed 40 key national climate adaptation and health policies of the focal countries in the South Asia region. It emerged from our textual analysis that 30 out of the 40 reviewed documents had a (human) disease focus. The remaining 5 documents had no specific disease focus. Of the 30 documents with human disease focus, 45.7% (n = 14 documents) had an explicit mention of specific human diseases of interest in the respective countries. Most the documents were general in their identification, lacking specificity on the individual climate-sensitive diseases of priority in national adaptation planning and health interventions (Table 1). As an exception, Bangladesh's Climate Change and Sustainable Development (2000–2050) specifically identifies cholera, diarrhoea, malaria, dengue and Lymphatic filariasis as the topmost diseases of policy interest. Similarly, India's National Action Plan on Climate Change (undated) identifies malaria, visceral leishmaniasis, Japanese encephalitis, Lymphatic filariasis and dengue as the priority diseases of focus. With the notable exception of the National Health Policy 2009 (mentioning avian influenza, malaria and dengue as priority diseases of focus), relevant documents reviewed for Pakistan generally lacked clarity on specific priority climate-and-water sensitive diseases of interest. The federal system of governance in Pakistan has meant that prioritisation of specific diseases is often reflected in policy/programme documents at the sub-national level. The priority diseases climate-and water sensitive diseases of interest in the South Asia region were again reflected in the number of mentions in the policy documents (malaria (n = 14 documents), dengue (n = 9 documents), Lymphatic filariasis (n = 6 documents) and cholera (n = 5 documents), as was the case with the peer reviewed literature. The amount of evidence existing for the above-mentioned diseases, in terms of disease system understanding and societal impacts, may have influenced their translation into policy as a function of their importance in the respective countries. If this holds true, then it stands to reason that the more evidence available on the burden and impact of a specific disease or pathogen, the greater its visibility in the national policy arenas.

## Priority climate-and water-sensitive diseases of interest

As evidenced in Fig 3A, most reviewed papers reported on malaria (n = 53 studies), Lymphatic filariasis (n = 46 studies), dengue (n = 41 studies) and cholera (n = 38 studies) as the topmost

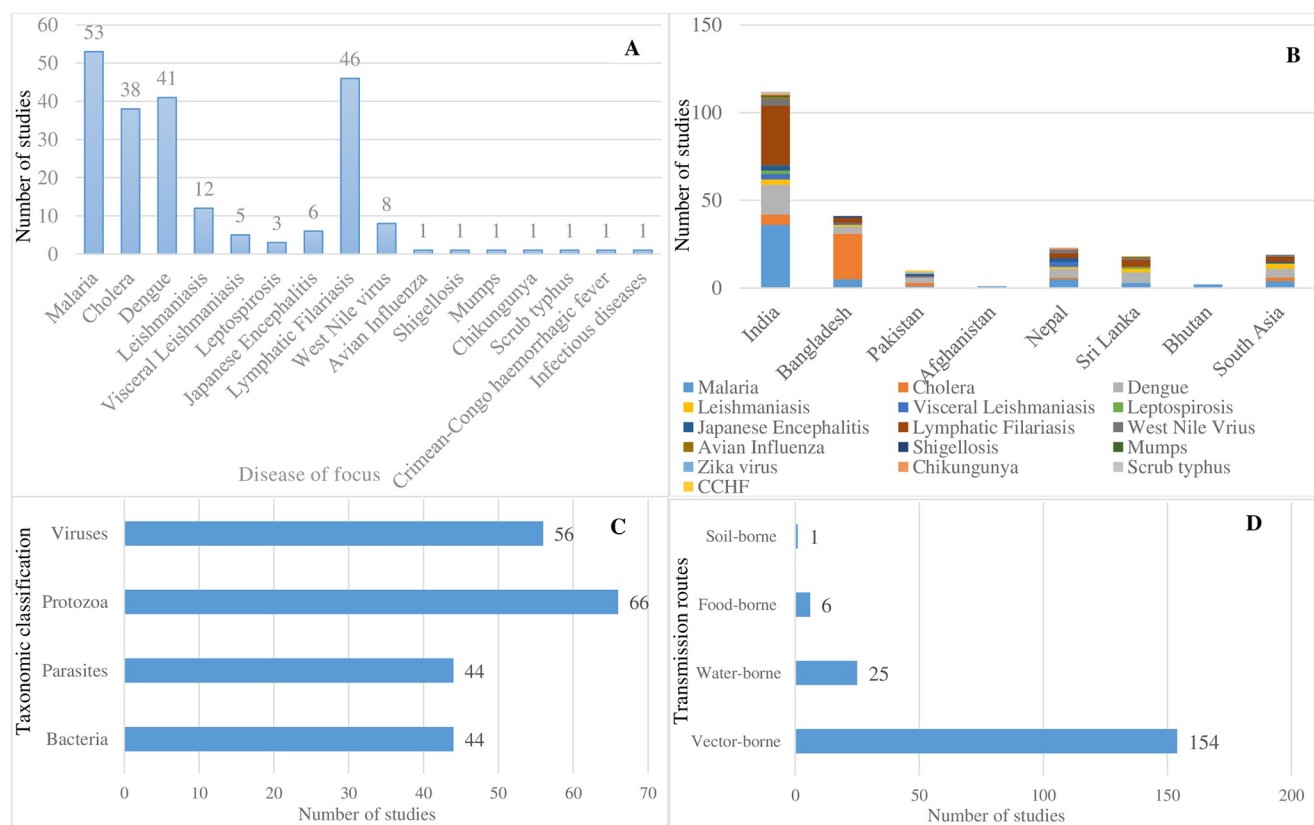

**Fig 3. Priority climate and water sensitive risks to human health.** Bar chart shows the number of papers by disease focus (A), disease and country of focus (B), taxonomic group of reported diseases (C), and disease transmission routes (D).

climate-and water sensitive diseases of research interest in the south Asia region. Of the 53 malaria focused studies, over two thirds of the papers (66.6% or n = 36 studies) had India as the focal country followed by 5 studies apiece conducted in Bangladesh and Nepal (Fig 3B). Papers related to Lymphatic filariasis were predominantly from India (n = 34 studies), Sri Lanka (n = 4 studies), Bangladesh (n = 3 studies) and Nepal (n = 3 studies) respectively. A significant proportion of the cholera-focussed studies were from Bangladesh (69.4% or n = 26 studies), reflecting the country's high-burden status (Fig 3B). For instance, it is estimated that 66 million people are at risk of cholera in Bangladesh, with approximately 100,000 cases and 4500 deaths annually [58–61].

Contrasting disease focus and publication trends (Figs 2A and 3B), the results show that most of the papers related to priority diseases were published within the last eight years alone (i.e., 2015 and 2023). This highlights the view that scientific research focus on climate-health theme is still nascent in the region, at least from the standpoint of publications in international refereed journals. A case in point is leptospirosis-focussed papers, which for example, had 23 of the 24 related studies published within the last seven years alone (i. e. 2015–2022). The more intense research focus may be reflective of the leptospirosis policy and programme in India which started in 2015. It is plausible that there is a longer history of operational research work in respective government departments/ institutes in the region that is not represented in the peer-reviewed literature, in which case further exploration through policy interviews would be instructive. On the taxonomic classification, the reviewed papers predominantly focussed on

protozoa (n = 66 studies), followed by viruses (n = 56 studies), parasites and bacteria (n = 44 studies respectively). Most of the papers reported on vector-borne diseases as the topmost climate-sensitive diseases in the region, which mirrors the common trend in the zoonoses scholarship, where most research continues to focus on climate change effects on the distribution and spread of vector-borne infectious diseases. Interestingly, the results show an under-representation of important, neglected climate-sensitive infectious vector-borne diseases (e.g. chikungunya and visceral leishmaniasis) in terms of scientific interest despite their wide-ranging societal impacts, particularly in the rural contexts of the South Asia region. Juxtaposing the disease focus of peer-reviewed papers and policy focus appear to give the indication that there is an overemphasis on known and epidemic climate-sensitive diseases (e.g. malaria, cholera) to the relative neglect of otherwise endemic high burden neglected diseases (e.g. leptospirosis, chikungunya) that disproportionately affect poor and vulnerable communities in the region.

## Environmental drivers and disease focus associations

To the extent that the dynamics of disease distribution and spread is likely to be affected by environmental change factors, it was instructive to identify the range of climate and non-climatic drivers considered in the reviewed studies. Of the studies that considered climate drivers (n = 109 papers), 59 (representing 54.1%) studies analysed 1 or 2 climate drivers with the remaining 50 studies reported on 3 or more drivers (Fig 4C). The climate drivers of disease (hydrometeorological hazards) considered in these studies included extreme weather events, seasonal rainfall, temperature, drought and wind. The most frequently addressed climate drivers were rainfall 65.1% (n = 71 studies), temperature 60.6% (n = 66 studies), hydrology (e.g.,

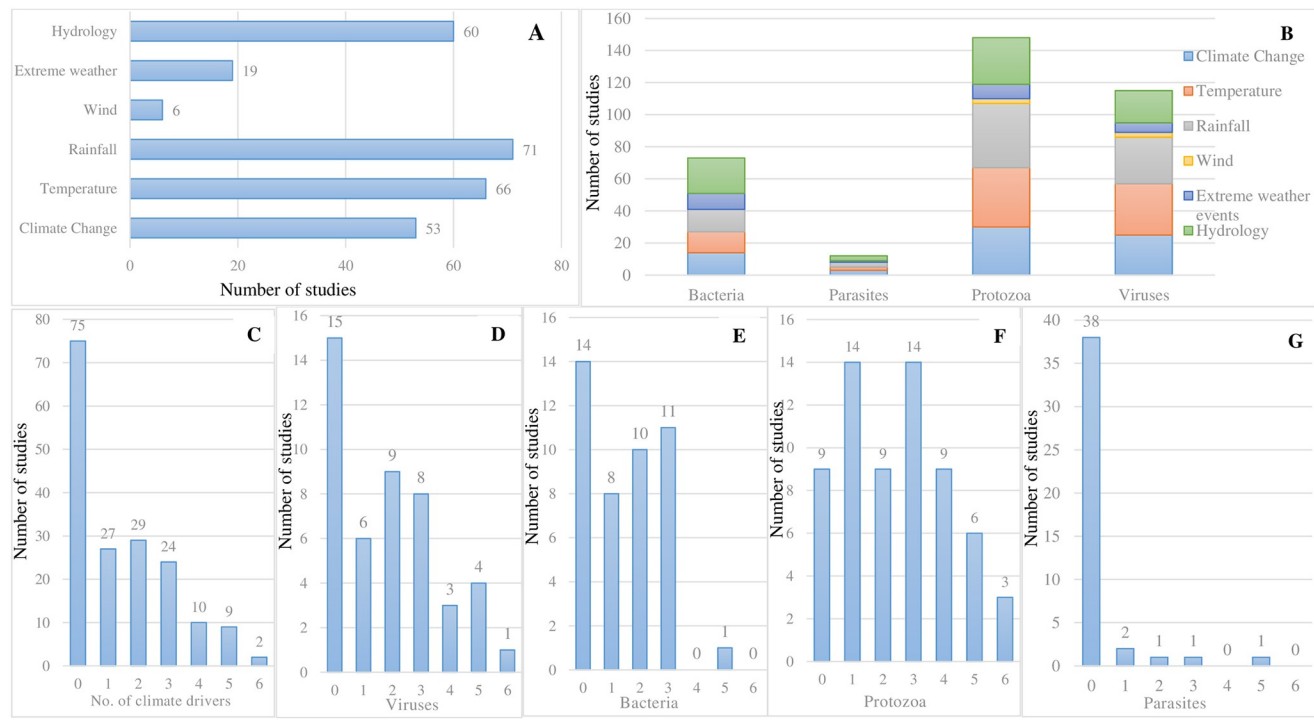

**Fig 4. Climate drivers of diseases reported in the reviewed studies.** (A) frequency histogram of the climate drivers considered in reviewed studies; (B) stacked bar chart showing the proportion to which focal diseases are sensitive to different climate drivers, broken down by disease taxa (note: More than one climate driver can be associated with a disease; (C) frequency of climate drivers reported; D, E, F & G represent frequency histograms of the number of climate drivers (0–6 drivers) considered in studies that focussed on viruses, bacteria, protozoa, and parasites, respectively.

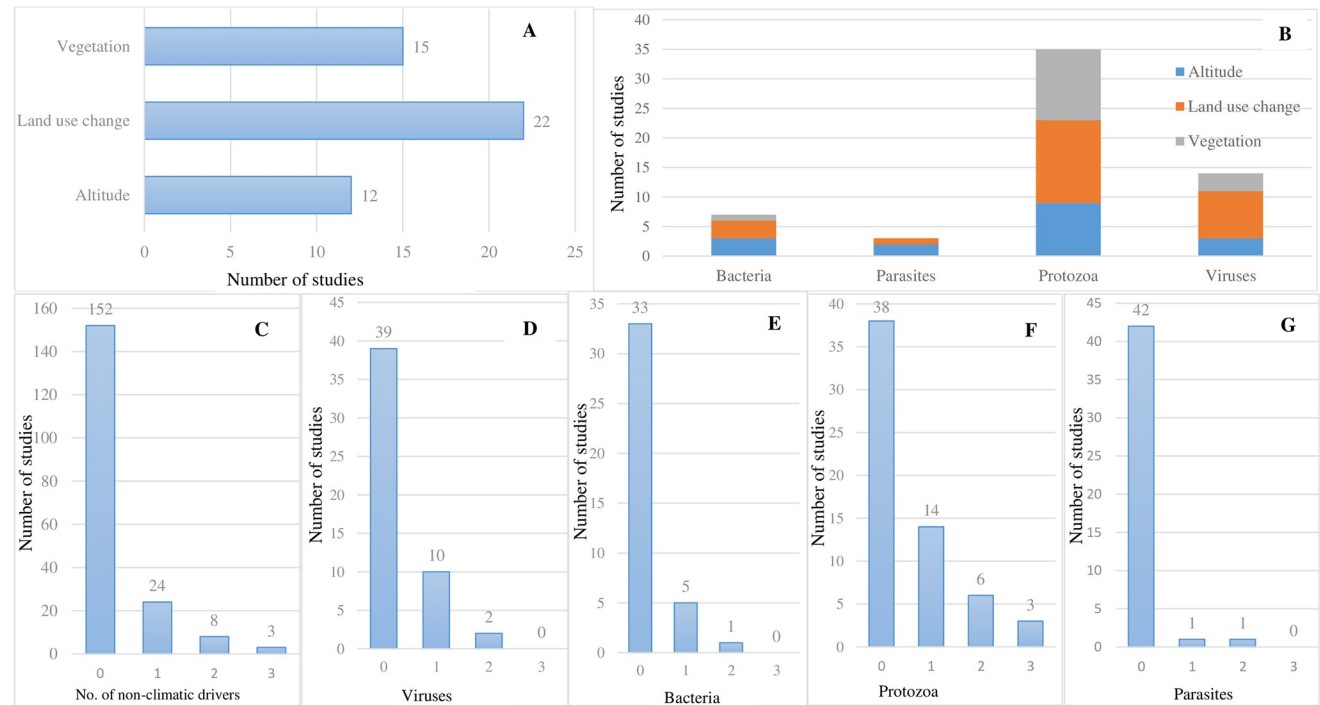

**Fig 5. Non-climatic drivers of diseases reported in the reviewed studies.** (A) frequency histogram of non-climatic drivers reported, (B) stacked bar chart showing the proportion to which focal diseases are sensitive to different non-climatic drivers, broken down by disease taxa (note: More than one non-climatic driver can be associated with a disease), (C) frequency histogram of number of non-climatic drivers considered in reviewed studies, D, E, F,& G frequency histograms of the number of non-climatic drivers (0–3 drivers) considered in studies that focused on viruses, bacteria, protozoa and parasites, respectively.

soil moisture, drought, flood extent and waterbodies) 55.1% (n = 60 studies) and climate change 48.6% (n = 53 studies) (Fig 4B). Extreme weather events (e.g., heatwaves, cyclones, and flash floods) 17.4% (n = 19 studies) and wind (mostly storm events) 5.5% (n = 6 studies) were the climate drivers least often examined as linked to focal diseases reported in respective studies (Fig 4B). The relatively limited focus on extreme weather events as a known climate driver of focal diseases reported, could be indicative of the dearth of research evidence on related drivers linked to disease burdens and/or transmission pathways in the region. It may also reflect the fact that it has been relatively easier for modellers to access input data on mean climatology than extreme events. Further cross-tabulation of climate drivers and disease investigated showed that the topmost diseases associated with seasonal rainfall pattern as a climate driver were malaria (n = 33 studies of 66), dengue (n = 28 studies) and cholera (n = 8 studies). Similarly, these diseases (malaria, dengue, and cholera) were frequently associated with temperature and climate change (see Fig 6).

For most studies of viruses, bacteria and protozoa, either 1 or no climate driver was considered within the study (Fig 4d–4g). The highest number of climate drivers (n = 6) considered for an individual disease was for malaria (protozoa, n = 2 studies), leishmaniasis (protozoa parasite, n = 2 studies), and dengue (viral, n = 2 studies), whereas Japanese encephalitis virus and avian influenza virus were each reported in single studies [62,63]. One cholera-related study had 5 associated climate drivers [22].

The non-climatic drivers examined in reviewed studies included altitude, vegetation, and land use change (n = 49 studies). Land use change and vegetation were the non-climatic

| | Climate Change | Temperature | Rainfall | vegetation | Hydrology* | Extreme weather events | Wind | Humidity | Land use | Migration | Population density | Poverty |
|---|---|---|---|---|---|---|---|---|---|---|---|---|
| Malaria | 27 | 30 | 33 | 9 | 26 | 7 | 2 | 13 | 13 | 8 | 4 | 2 |
| Lymphatic Filariasis | 4 | 4 | 6 | 0 | 3 | 1 | 0 | 2 | 0 | 0 | 0 | 4 |
| Dengue | 22 | 26 | 28 | 4 | 17 | 7 | 4 | 10 | 8 | 2 | 6 | 2 |
| Cholera | 11 | 7 | 8 | 1 | 21 | 6 | 0 | 3 | 3 | 1 | 3 | 2 |
| Leishmaniasis | 6 | 9 | 11 | 2 | 5 | 5 | 1 | 4 | 3 | 0 | 1 | 1 |
| West Nile virus | 0 | 1 | 1 | 0 | 0 | 1 | 0 | 0 | 0 | 0 | 0 | 0 |
| Japanese Encephalitis | 3 | 3 | 3 | 0 | 2 | 2 | 1 | 1 | 1 | 1 | 0 | 0 |
| Visceral Leishmaniasis | 1 | 0 | 2 | 1 | 0 | 0 | 1 | 0 | 0 | 0 | 0 | 0 |
| Leptospirosis | 0 | 1 | 2 | 0 | 2 | 0 | 0 | 1 | 1 | 0 | 1 | 0 |
| Scrub typhus | 0 | 0 | 0 | 0 | 0 | 0 | 0 | 0 | 0 | 0 | 0 | 0 |
| Crimean-Congo haemorrhagic fever | 0 | 1 | 0 | 0 | 0 | 0 | 0 | 0 | 0 | 0 | 0 | 1 |

**Fig 6. Cross-tabulation of assignment of papers to a given environmental and/or social driver and disease lens in the South Asia region.** Displayed are selected environmental change and socio-demographic drivers (mentioned more than 5 times across the reviewed papers) in a matrix of disease lens in each reviewed paper. Disease lens and climate and non-climatic drivers are sorted to frequency of inclusion/consideration in respective studies (i.e., studies from the general climate change and rainfall driver lens respectively are most frequently focussed on malaria and dengue). Note: EWEs = Extreme Weather Events; * = soil moisture, drought, flood & waterbodies.

drivers most often considered in studies that considered these types of drivers (Fig 5A). Cross-tabulation of non-climatic drivers and disease investigated revealed that malaria (n = 12 studies of 22), dengue (n = 8 studies) and cholera (n = 3 studies) were the topmost diseases associated with land use change as a non-climate driver (also see Fig 6). The highest proportion of non-climatic drivers (n = 3) addressed in respective studies was for protozoan-related diseases (malaria) (Fig 5F).

Only 34 studies (18.2%) considered both climate and non-climatic environmental drivers, out of which 14 were modelling focussed. While the limited consideration of both climate and non-climatic drivers within studies is suggestive of data gaps on specific drivers, it also underscores the relevance of cross-sectoral information exchange, for example from hydroclimate and health, in better prioritising and planning of appropriate disease control interventions (see Section 3.4). Models describing the impact of climate change on infectious diseases considering a narrow range of climate drivers known to affect diseases may risk large uncertainties in comprehensively understanding and projecting how climate change will affect these same diseases in the future. This lends support to McIntyre et al. [55] observation about the urgent need for disease risk models, to examine a combination of climate drivers and multiple scenarios for changes in climate impacts [64].

A small proportion of all the reviewed studies (28.3% or n = 53 studies) identified socio-demographic factors as drivers of disease emergence or spread. For example, the studies that reported on social drivers of disease transmission focussed on malaria (44.4% or n = 20

studies), dengue (33.3% or n = 15 studies), cholera (17.8% or n = 8 studies) and Lymphatic filariasis (13.3% or n = 6 studies). Of the 45 studies that considered social drivers, 73.3% of them (n = 33 studies) also included climate and 46.7% (n = 21 studies) non-climatic drivers. The limited focus on the interplay of socio-demographic factors and disease dynamics (underscored by the difficulty of obtaining socio-demographic data at the same resolution as health data) perhaps highlights a strategic research priority for better understanding the biosocial factors that shape vulnerability to climate-sensitive disease risks in the region.

### Forecasting capacities to address climate-driven infectious diseases transmission and spread: Taking stock

To better understand the forecasting capacities, priorities and needs for targeted early warning and anticipatory response, it was imperative to investigate the diversity of existing national hydroclimate and disease surveillance systems available to stakeholders in the region, including sectoral focus, scale of operationalisation and priority diseases being addressed. From the analysis of the reviewed papers, only 36 studies reported on national/state level health surveillance or information systems, with just 3 studies mentioning the existence of integrated surveillance systems (i.e. linking animal and human health information) for disease prevention and control [65–67]. None of the reviewed studies reported on linkage of health and environmental information systems to inform disease management in the reviewed countries. This indicates limited information systems in the reviewed countries, possibly reflecting the relative dearth of evidence on forecasting systems. It therefore follows that further investigation through targeted key informant interviews and workshops remains necessary to better understand and assess forecasting capabilities and opportunities to link information and models across sectors.

The evidence for existing hydroclimate and disease surveillance capacities in the reviewed countries is summarised in Fig 7. To better understand the needs for hydroclimate surveillance for different diseases, we used identified links between climatic drivers (different

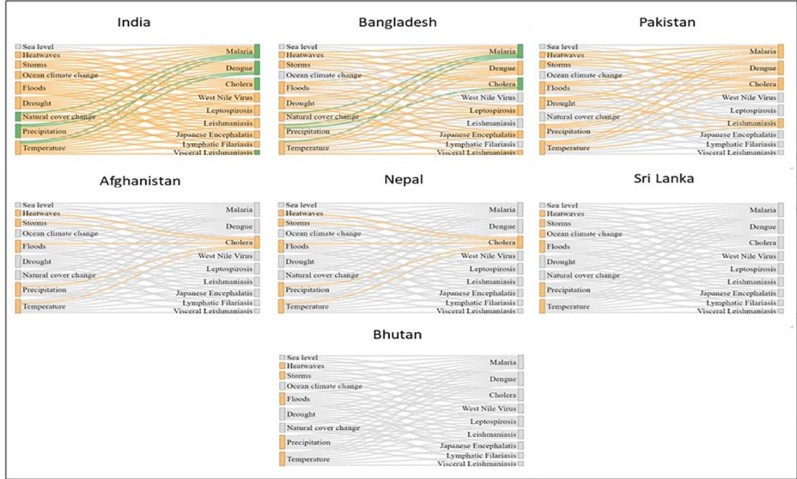

**Fig 7. Summary of existing hydroclimate and disease surveillance capacities.** Panels show all potential drivers (hydrometeorological hazards) and links with priority diseases obtained from Mora et al. [10] in grey. Orange nodes indicate that information is available for a driver or disease from existing information systems and orange links highlight where information may be available to explore the relationship between a driver and disease. Green links highlight instances where a link between a driver and disease was explored in a reviewed study and green nodes highlight instances where national information systems have fed into a reviewed study.

hydrometeorological hazards) and diseases outlined in Mora et al [10] with further detailed results available in S3–S6 Tables. Whereas some of the identified information systems have yet to be fully operationalised and thus are not mature to assess in terms of impact, they present a promising outlook and a fair representation of the existing capabilities and avenues for improvement, particularly in terms of strategic emphasis, sectoral focus and diseases being prioritised for interventions (S3–S6 Tables). As evidenced in Fig 7, our inventory of existing information systems indicate that the largest number of hydroclimate and health information systems are found in India (n = 18 platforms), followed by Bangladesh (n = 4 platforms) and Pakistan (n = 3 platforms), which somewhat reflects their socio-economic capacity (as upper middle-income nations) and investment priorities. Afghanistan and Sri Lanka had the least number of information systems (1 apiece). In Afghanistan, the prevailing political tensions, which has gravely impacted on national infrastructural investments including health and metrological systems [61,68–70], may provide part of the explanation for this lack of evidence. Besides, the restricted composition of the WHO South Asia region (excluding Afghanistan and Pakistan) may impact on broader regional cooperation in terms of information exchange and capacity building. Moreover, almost all the identified information systems (91.2% or n = 31 platforms) had national orientation and a seemingly limited disease focus, suggesting that few of the countries have their own surveillance systems to monitor and track the outbreak of any new diseases.

In terms of cross-sectoral implementation, a large majority of the information systems are sectorally driven in terms of operationalisation, which poses a risk for late detection and cross-sectoral engagement with preparedness and response infrastructure. To the extent that most of the identified hydro-climate systems utilise hydro-metrological data highlights a potential for use in establishing relationships with health surveillance and information systems, where corresponding disease data exist. In this sense, it is imperative that hydro-climate forecasting outputs are available on timescales relevant to disease surveillance. Altogether, the foregoing highlights varying capabilities and sectoral representation, underscoring the need for crosscutting integrated efforts (through the development of cross-sectoral early warning systems and tools) to tackle risks and vulnerabilities as well as support long term adaptation planning, particularly in highly vulnerable countries like Sri Lanka, Bhutan and Bangladesh.

## Developing climate-driven early warning systems to support disease adaptation and resilience building interventions—A research agenda

Hydro-climatic and disease information systems are increasingly recognised as vital for successful adaptation in the public health sectors in the face of climate change, through providing timely and tailored information to support targeted decision-making and resilience building interventions in highly vulnerable regions [49,71–73]. While conventional sectorally-driven hydrological, climate and disease early warning systems are useful, they remain inadequate for comprehensive capture of complex disease dynamics that straddle the human, animal, and environment interface. This necessitates cross-sectoral efforts to develop integrated climate-driven early warning systems that leverage existing meteorological and health surveillance capacities across scale. Yet, much of the research and efforts toward improving the delivery and use of hydroclimatic information and disease surveillance capabilities in the South Asia region has at best been fragmented and sectorally focussed [66,73–77]. The operationalisation of the One Health approach (which recognises the interconnectedness of human health, animal health and the environment, and advocates for integrated, holistic and transdisciplinary approaches to reduce disease impacts) is seen as a plausible vehicle for developing integrated solutions that support disease control interventions at scale [20,50,78].

In this paper, we considered the question of what is needed to develop integrated climate-driven early warning systems for climate-and water-sensitive diseases to support climate-health, health adaptation, and risk management needs at scale. Consistent with the notion that cross-sectoral collaboration is needed to better inform effective climate-health interventions, we argue that there is a need for a paradigm shift in research focus from a single pathogen/disease focus to multiple disease stressors, along with their burdens and impacts, to support the development and targeting of early warning systems. To our knowledge, this scoping review is the first of its kind to characterise the state of the art on climate-health scholarship in the South Asia region, highlighting the priority climate-and water-sensitive diseases and existing cross-sectoral information systems that could be linked in the future to better understand and predict how climate variability and climate change precipitate health impacts. We found that there exist marked differences in research foci in the region and existing information systems capabilities, with overrepresentation of evidence from India and Bangladesh compared to the lesser developed and highly vulnerable countries (Afghanistan, Sri Lanka and Bhutan), which are lagging in terms of research, and hydrological and surveillance infrastructural capacity. In Afghanistan, for example, research evidence on climate-health linkages is largely focussed on vector-borne diseases (malaria and dengue) [21,61,69,79,80], despite the potential climate change impacts on cholera, which is a priority epidemic disease of concern (128,278 cases reported between 2000 and 2017) in the country [81].

We thus put forward five directions for future research and operationalisation.

1. To guarantee more robust evidence on disease burdens and impacts, we need further research on priority climate-and water sensitive diseases particularly in least represented and data poor contexts such as Afghanistan, Sri Lanka, and Bhutan. As evidenced above, there is a disproportionate representation of research focus on India and Bangladesh which has meant a dearth of evidence on other countries to support long-term adaptation planning. Nonetheless, it is noteworthy that poor national security in some of the focal countries (e.g., Afghanistan) hampers the ability to undertake fieldwork to assess adaptation options and requirements [82].

2. Studies focussing on the social determinants of climate impacts on health are underrepresented among the topic areas in the literature on the South Asia region. Given that patterns of vulnerability and adaptation to climate induced disease risks and associated impacts manifests within dynamic socio-cultural and political contexts, it is critical to understand the context of at–risk populations, drivers of their vulnerability, and adaptive strategies to better inform the contextualisation and tailoring of early warning support systems for bolstering resilience and long-term adaptation. This could be delivery of fine scale vulnerability maps and risk information that integrate understanding of social vulnerability and improves targeting of interventions.

3. Better understanding of the different information needs, opportunities, and barriers for cross-sector information-sharing and analytics is needed across the region to support the development of locally appropriate climate-driven early warning systems for meaningful adaptation planning at the relevant scales. The paucity of studies assessing cross-sectoral forecasting and surveillance capacities and requirements calls for better inclusion of underrepresented countries (Afghanistan, Pakistan, Sri Lanka, and Bhutan) in future research. International funding support to address these questions, particularly in the underrepresented countries, and strengthening regional research networks could be a good starting point.

4. The non-accessibility of data and/or barriers to information-sharing between cross-sectoral actors remains a key challenge, as data is treated as proprietary. This calls for greater

sensitisation of data owners and curators, and disciplinary experts in climate and epidemiology, about cross-sectoral information-sharing and improved mechanisms for fostering these and needs for developing integrated early-warning systems and decision support tools to better inform disease management and long-term adaptation planning, learning from demonstration cases in other contexts.

5. There is also a need to create improved information services in which producers and end-users of early warning information and decision-support tools interact to identify needs and priorities for reducing vulnerabilities and strengthening resilience to climate induce disease risks [50,83]. This suggests a need-based tailored information service for at-risk populations in the region. In doing so, co-production approaches remain an important aspect of tailoring hydroclimatic and disease information systems to stakeholder needs and priorities.

Notwithstanding the breadth and the wide regional coverage of the evidence reviewed, this scoping review is subject to some limitations. First, scoping reviews have an inherent limitation due to the focus on the breadth rather than depth of information on the subject-matter. As such, meta-analysis of the evidence base (including quality appraisal) is generally not conducted in a scoping review [84]. Second, owing to time and COVID-19-related constraints, it was not possible to engage directly with stakeholders about their needs and priorities as well as to ascertain capabilities and data access to enable a joined-up approach through key informant interviews or multi-stakeholder workshops. Third, the reliance on English-language peer-reviewed papers and national climate adaptation plans and health policies might have resulted in missing some relevant policy documents (e.g. regional/state policies) and papers published in restricted local journals and/or local languages. Moreover, although we focussed on policies current at the time of review, we cannot dismiss the possibility that we missed new documents or that a policy was updated before the review was completed. This scoping review nonetheless showcases some critical patterns in existing climate-health scholarship and related forecasting and surveillance systems in South Asia which have resulted in a number of key actionable steps for developing integrated climate-driven early warning infrastructure towards supporting long-term adaptation planning and interventions.

## Supporting information

**S1 Checklist. PRISMA checklist.**
(DOCX)

**S1 Table. Summary characteristics of reviewed studies.**
(DOCX)

**S2 Table. Annotated key studies reviewed by focal diseases of interest.**
(DOCX)

**S3 Table. Overview of the main hydroclimate and health information systems and programs in the focal countries in south Asia.**
(DOCX)

**S4 Table. Overview of forecasting ability of National Meteorological Departments.**
(DOCX)

**S5 Table. Overview of the health information systems and programs in the focal countries in south Asia.**
(DOCX)

**S6 Table. Summary of studies where links were explored between driver and disease or national level health information systems feed into a study.**
(DOCX)

**S7 Table. Summary of all studies identified in the literature search.**
(DOCX)

**S1 File. Exemplar search strings.**
(DOCX)

**S2 File. Risk of bias assessment.**
(XLSX)

**S3 File. Scoring database.**
(XLSX)

**S1 Graphical abstract.**
(PPTX)

## Author Contributions

**Conceptualization:** Festus A. Asaaga, Sunita Sarkar, Bethan V. Purse.

**Data curation:** Emmanuel S. Tomude, Nathan J. Rickards.

**Formal analysis:** Festus A. Asaaga, Emmanuel S. Tomude, Nathan J. Rickards, Richard Hassall, Bethan V. Purse.

**Funding acquisition:** Festus A. Asaaga, Bethan V. Purse.

**Investigation:** Festus A. Asaaga, Emmanuel S. Tomude, Sunita Sarkar, Bethan V. Purse.

**Methodology:** Festus A. Asaaga, Emmanuel S. Tomude, Richard Hassall, Bethan V. Purse.

**Project administration:** Festus A. Asaaga, Sunita Sarkar, Bethan V. Purse.

**Software:** Festus A. Asaaga, Emmanuel S. Tomude, Richard Hassall.

**Supervision:** Festus A. Asaaga, Sunita Sarkar, Bethan V. Purse.

**Visualization:** Festus A. Asaaga, Emmanuel S. Tomude, Richard Hassall.

**Writing – original draft:** Festus A. Asaaga, Emmanuel S. Tomude.

**Writing – review & editing:** Festus A. Asaaga, Emmanuel S. Tomude, Nathan J. Rickards, Richard Hassall, Sunita Sarkar, Bethan V. Purse.

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
