## [Decision Letter · Decision Letter 0]

30 Apr 2024

PONE-D-24-09817

Informing climate-health adaptation options through mapping the needs and potential for integrated climate-driven early warning forecasting systems in South Asia

PLOS ONE

Dear Dr. Asaaga,

Thank you for submitting your manuscript to PLOS ONE. After careful consideration, we feel that it has merit but does not fully meet PLOS ONE’s publication criteria as it currently stands. Therefore, we invite you to submit a revised version of the manuscript that addresses the points raised during the review process.

We look forward to receiving your revised manuscript.

Kind regards,

Cristiana Abbafati, Ph.D.

Academic Editor

PLOS ONE

Journal Requirements:

2. During your revisions, please revise your title to specify that your study is a Scoping Review, and update it in the manuscript file and online submission information if needed.

   "The research was funded by NERC as part of National Capability"

6. Please upload a copy of Figure 8, to which you refer in your text on page 29. If the figure is no longer to be included as part of the submission please remove all reference to it within the text.

Additional Editor Comments:

Reviewer 1

Several methodological issues that need to be addressed before considering it for publication.

Please, find here my comments and suggestions.

The abstract is quite verbose, especially for the results section. In this regard, my suggestion is to retrieve and summarize the main data from the corresponding section of the main text.

There is a slight misperception about what constitutes an emerging disease versus simply an outbreak of a non-emerging disease. My suggestion is that someone with some experience in infectious diseases, their transmission, and potential vectors (which can also be influenced by climate), reviews the text.

I don't understand why the work is limited to South Asia; perhaps the introduction should better explain the rationale.

Lines 131-133 could be removed.

Line 156. The methodology used for this additional research (or second stage of research, as the authors prefer to identify it) is not very clear. Were free searches used? Keywords? Clarification would be helpful.

Line 276, the name of journal in which the articles included in the review are published is not relevant information. Moreover, it is also excluded from the classical methodology for conducting systematic and scoping reviews.

Figure 2 is only partially informative.

The authors should replace it with a table summarizing the "typical" characteristics of studies included in systematic reviews: first author, year of publication, country, study design, main findings, possible bias, funding, and quality of that evidence.

Additionally, for this last point, I do not see the choice of a specific tool for assessing the quality of that evidence. It is important to remember that a systematic review may be undertaken to confirm or refute whether current practice is based on relevant evidence, to establish the quality of that evidence, and to address any uncertainty or variation in practice that may be occurring.

Reviewer 2

I suggest authors mention the type of study (ie., scoping review) in the title of the article.

Introduction

I suggest verifying the correct use of the term “hazard” (eg., lines 53, 56) by consulting the report summarising extensive work on terminology and definitions carried out by the DRR scientific community (https://www.undrr.org/publication/hazard-definition-and-classification-review-technical-report).

Methods

I cannot find Page et al., 2020 (line 138) in the reference list at the bottom of the manuscript. In this regard, I would suggest considering/citing the PRISMA Extension for Scoping Reviews (PRISMA-ScR) checklist.

I don’t think it is necessary to include a table with your search strings in the methods, it can go in the Supporting information, Appendix.

In section 2.2 I would have appreciated reading a few lines about your operational definitions of core concepts. This helps understanding how you applied eligibility criteria and enhances replicability of your protocol.

Text in lines 185-187 and the PRISMA flowchart are better suited to the Results section.

Results & Discussion

Overall, authors have clearly reported and critically reflected upon results, and I was pleased to see integration of findings from scientific articles and policy documents.

I believe that in Section 3.3 there is again confusion regarding terminology, as some of the so-called hazards (eg., extreme weather events) are referred to as climate drivers. I am aware that there might be vocabulary-related discrepancies between the climate science community and the disaster science community. Perhaps, for the sake of clarity, it would be good to cite sources of definitions. Reporting operational definitions in the methods, as suggested above, might help.

I was wondering whether “national/state level health surveillance or information systems” (lines 476-477) is limited to systems and mechanisms implemented by the state, or whether you adopted a broader definition of health system that also included non-state actors such as international humanitarian organization, possibly relevant for countries like Afghanistan. Perhaps a reference to this issue might be of interest in section 3.4?

General comments

I believe only considering articles - esp. local policies - written in English is a very important limitation. If it is not possible to include documentation written in the local languages, I would advise specifying where possible the number of records that were excluded for language reasons and in any case to reflect on possible selection bias, and implications, in the results/discussion.

Reviewers' comments:

Reviewer's Responses to Questions

**Comments to the Author**

1. Is the manuscript technically sound, and do the data support the conclusions?

Reviewer #1: Yes

Reviewer #2: No

2. Has the statistical analysis been performed appropriately and rigorously? 

Reviewer #1: Yes

Reviewer #2: No

3. Have the authors made all data underlying the findings in their manuscript fully available?

Reviewer #1: Yes

Reviewer #2: No

4. Is the manuscript presented in an intelligible fashion and written in standard English?

Reviewer #1: Yes

Reviewer #2: Yes

5. Review Comments to the Author

Reviewer #1: Dear authors,

I appreciated reading your manuscript titled: “Informing climate-health adaptation options through mapping the needs and potential for integrated climate-driven early warning forecasting systems in South Asia”. Please, find some suggestions below on how to improve the manuscript for publication in PLOS One.

Title & Abstract

I suggest authors mention the type of study (ie., scoping review) in the title of the article.

Introduction

I suggest verifying the correct use of the term “hazard” (eg., lines 53, 56) by consulting the report summarising extensive work on terminology and definitions carried out by the DRR scientific community (https://www.undrr.org/publication/hazard-definition-and-classification-review-technical-report).

Methods

I cannot find Page et al., 2020 (line 138) in the reference list at the bottom of the manuscript. In this regard, I would suggest considering/citing the PRISMA Extension for Scoping Reviews (PRISMA-ScR) checklist.

I don’t think it is necessary to include a table with your search strings in the methods, it can go in the Supporting information, Appendix.

In section 2.2 I would have appreciated reading a few lines about your operational definitions of core concepts. This helps understanding how you applied eligibility criteria and enhances replicability of your protocol.

Text in lines 185-187 and the PRISMA flowchart are better suited to the Results section.

Results & Discussion

Overall, authors have clearly reported and critically reflected upon results, and I was pleased to see integration of findings from scientific articles and policy documents.

I believe that in Section 3.3 there is again confusion regarding terminology, as some of the so-called hazards (eg., extreme weather events) are referred to as climate drivers. I am aware that there might be vocabulary-related discrepancies between the climate science community and the disaster science community. Perhaps, for the sake of clarity, it would be good to cite sources of definitions. Reporting operational definitions in the methods, as suggested above, might help.

I was wondering whether “national/state level health surveillance or information systems” (lines 476-477) is limited to systems and mechanisms implemented by the state, or whether you adopted a broader definition of health system that also included non-state actors such as international humanitarian organization, possibly relevant for countries like Afghanistan. Perhaps a reference to this issue might be of interest in section 3.4?

General comments

I believe only considering articles - esp. local policies - written in English is a very important limitation. If it is not possible to include documentation written in the local languages, I would advise specifying where possible the number of records that were excluded for language reasons and in any case to reflect on possible selection bias, and implications, in the results/discussion.

Reviewer #2: I want to thank the Editor for the opportunity of reading and reviewing the work by Asaaga and coll, titled “Informing climate-health adaptation options through mapping the needs and potential for integrated climate-driven early warning forecasting systems in South Asia”. The aim of this scoping review was to delineate and outline priority climate- and water-sensitive diseases, assess existing forecasting and surveillance systems within climate and health sectors, and explore the requirements and opportunities for developing integrated climate-driven early warning forecasting systems to aid long-term adaptation planning and interventions in the South Asia region.

Overall, the work is well written, but contains several methodological issues that need to be addressed before considering it for publication.

Please, find here my comments and suggestions.

The abstract is quite verbose, especially for the results section. In this regard, my suggestion is to retrieve and summarize the main data from the corresponding section of the main text.

There is a slight misperception about what constitutes an emerging disease versus simply an outbreak of a non-emerging disease. My suggestion is that someone with some experience in infectious diseases, their transmission, and potential vectors (which can also be influenced by climate), reviews the text.

I don't understand why the work is limited to South Asia; perhaps the introduction should better explain the rationale.

Lines 131-133 could be removed.

Line 156. The methodology used for this additional research (or second stage of research, as the authors prefer to identify it) is not very clear. Were free searches used? Keywords? Clarification would be helpful.

Line 276, the name of journal in which the articles included in the review are published is not relevant information. Moreover, it is also excluded from the classical methodology for conducting systematic and scoping reviews.

Figure 2 is only partially informative.

The authors should replace it with a table summarizing the "typical" characteristics of studies included in systematic reviews: first author, year of publication, country, study design, main findings, possible bias, funding, and quality of that evidence.

Additionally, for this last point, I do not see the choice of a specific tool for assessing the quality of that evidence. It is important to remember that a systematic review may be undertaken to confirm or refute whether current practice is based on relevant evidence, to establish the quality of that evidence, and to address any uncertainty or variation in practice that may be occurring.

6. PLOS authors have the option to publish the peer review history of their article (what does this mean?). If published, this will include your full peer review and any attached files.

Reviewer #1: No

Reviewer #2: No

---

## [Author Response · Author response to Decision Letter 0]

21 Jun 2024

Response to Editorial and Reviewers’ comments on manuscript – PONE-D-24-09817 - Informing climate-health adaptation options through mapping the needs and potential for integrated climate-driven early warning forecasting systems in South Asia

Overview of changes made

Very many thanks to the Reviewers and Editor for their time and effort in reviewing our paper. We have made changes to the paper in response to the editorial and reviewers’ comments. Please see in italics our detailed response to comments:

Editorial comments:

Response: Thanks for the comment. We have formatted the revised manuscript to conform with PLOS ONE style requirements. 

2. During your revisions, please revise your title to specify that your study is a Scoping Review, and update it in the manuscript file and online submission information if needed.

Response: We have revised the manuscript title to include scoping review as per the editorial and reviewer #2 suggestion. 

3. Thank you for stating the following financial disclosure: "The research was funded by NERC as part of National Capability". Please state what role the funders took in the study. If the funders had no role, please state: "The funders had no role in study design, data collection and analysis, decision to publish, or preparation of the manuscript." If this statement is not correct you must amend it as needed.

Response: Thanks for the comment. We have clarified in the financial disclosure statement that the funders had no role in study design, data collection and analysis, decision to publish, or preparation of the manuscript.

Response: Apologies for the confusion. We have removed the said phrase “data not shown” in the revised manuscript.

5. Please upload a copy of Figure 8, to which you refer in your text on page 29. If the figure is no longer to be included as part of the submission please remove all reference to it within the text.

Response: This is a typographical error and should have read Figure 7 instead of Figure 8 as captured in the original manuscript. We have replaced Figure 8 with Figure 7 in the revised manuscript (see line 458, page 27 of the revised manuscript). 

6.Please include captions for your Supporting Information files at the end of your manuscript, and update any in-text citations to match accordingly. Please see our Supporting Information guidelines for more information: http://journals.plos.org/plosone/s/supporting-information.

Response: We have included the captions of the Supporting Information files at the end of the revised manuscript and updated in-text citations as per the editorial comment. 

Reviewer #1:

1. The abstract is quite verbose, especially for the results section. In this regard, my suggestion is to retrieve and summarize the main data from the corresponding section of the main text.

Response: We have reduced the abstract, particularly in the methodological section, to focus on the results and implications.

2. There is a slight misperception about what constitutes an emerging disease versus simply an outbreak of a non-emerging disease. My suggestion is that someone with some experience in infectious diseases, their transmission, and potential vectors (which can also be influenced by climate), reviews the text.

Response: Thanks for the comment. We have reviewed the text and cannot find instances of where this misperception appears

3. I don't understand why the work is limited to South Asia; perhaps the introduction should better explain the rationale.

Response: Thanks for the comment. Apologies if the reviewer missed the justification in the original manuscript. We feel that rationale for delimiting the review to South Asia has been explained in the introduction of the original manuscript. For context, South Asia is widely recognised as one of the climate vulnerable regions globally and plagued with a considerable proportion of the global burden of infectious diseases. The region is also home to quarter of the world’s population. Nonetheless, we have added an additional reference of Intergovernmental Panel on Climate Change (IPCC) seminal publication (https://www.ipcc.ch/site/assets/uploads/2018/02/ar4-wg2-chapter10-2.pdf) speaking the vulnerability of the region from a climate and infectious disease perspective which we feel should suffice as justification for the study focus (see lines 73-107 of the revised manuscript). 

4. Lines 131-133 could be removed.

Response: We have removed lines 131-133 as suggested by the reviewer. 

5. Line 156. The methodology used for this additional research (or second stage of research, as the authors prefer to identify it) is not very clear. Were free searches used? Keywords? Clarification would be helpful.

Response: Apologies for the confusion. Following the literature search, we conducted an online search of relevant national climate adaptation plans, health policies and hydroclimate and health surveillance systems of the focal countries. We used the keywords “climate policy”, “health policy or legislation”, “climate change adaptation policy”, “hydroclimate information systems”, “health surveillance systems” and “country” were used to search documents about policies and legislations of relevance to the subject-matter (see lines 149-152, page 7 of the revised manuscript). 

6. Line 276, the name of journal in which the articles included in the review are published is not relevant information. Moreover, it is also excluded from the classical methodology for conducting systematic and scoping reviews.

Response: Thanks for the suggestion. Although we appreciate the context of the reviewer’s observation, we feel that for the including the name of journal in which included articles in the review are publish gives context to the literature landscape on the subject-matter at least for international readership. In any event, it affords some insight as to where relevant scholarship on the topic (i.e. climate-health studies on South Asia) are frequently published as we argue for more ‘representational’ studies on the region. 

7. Figure 2 is only partially informative. The authors should replace it with a table summarizing the "typical" characteristics of studies included in systematic reviews: first author, year of publication, country, study design, main findings, possible bias, funding, and quality of that evidence.

Response: Apologies if the reviewer missed this. We previously provided the “typical characteristics of included studies as supplementary information in the original manuscript submission (see S1 Table. Summary characteristics of reviewed studies as part of the supporting information). 

8. Additionally, for this last point, I do not see the choice of a specific tool for assessing the quality of that evidence. It is important to remember that a systematic review may be undertaken to confirm or refute whether current practice is based on relevant evidence, to establish the quality of that evidence, and to address any uncertainty or variation in practice that may be occurring.

Response: Thanks for the comment. As the study is a Scoping Review it is noteworthy that quality appraisal of the evidence is not mandatory. Indeed, the purpose of a scoping review is to explore the breadth of available evidence whilst systematic reviews on the other hand, aim to address very specific question and not exploratory. In the case of the latter a quality appraisal as the reviewer right alludes to is essential (see, e.g. Song et al. 2021 - https://doi.org/10.1371/journal.pone.0251440). 

Reviewer #2:

1. I suggest authors mention the type of study (ie., scoping review) in the title of the article.

Response: Thanks for the suggestion. We have revised the title of the paper as per the reviewer’s comment (see lines 1-3 of the revised manuscript). The revised title now reads:

Informing climate-health adaptation options through mapping the needs and potential for integrated climate-driven early warning forecasting systems in South Asia – a scoping review 

Introduction

2. I suggest verifying the correct use of the term “hazard” (eg., lines 53, 56) by consulting the report summarising extensive work on terminology and definitions carried out by the DRR scientific community (https://www.undrr.org/publication/hazard-definition-and-classification-review-technical-report). 

Response: Thank you for this insightful comment. This confusion has arisen due to disciplinary differences in what constitutes hazard between hydrometeorology and biological, infectious disease fields. We have removed reference in the introduction to climate-related hazards from diseases, since this could be confused with the use of hazard in hydrometeorology, which means weather-related, hydrometeorological events which can cause harm to humans, property, livelihoods, resources, and the environment. Instead we use the term climate-related risks.

Methods

3. I cannot find Page et al., 2020 (line 138) in the reference list at the bottom of the manuscript. In this regard, I would suggest considering/citing the PRISMA Extension for Scoping Reviews (PRISMA-ScR) checklist.

Response: We have added Page et al. 2021 to the reference list as suggested by the reviewer (see lines 755-758, page 41 of the revised manuscript). Additionally, we added a PRISMA Extension for Scoping Reviews (PRISMA-ScR) checklist as supporting information in the original manuscript submission – apologies if the reviewer missed this (see also S1 Checklist. PRISMA checklist as part of the supporting information).

4. I don’t think it is necessary to include a table with your search strings in the methods, it can go in the Supporting information, Appendix.

Response: We have moved the table with the search strings to Supporting information as per the reviewer suggestion (see S1 File. Exemplar search strings as part of the supporting information). 

5. In section 2.2 I would have appreciated reading a few lines about your operational definitions of core concepts. This helps understanding how you applied eligibility criteria and enhances replicability of your protocol.

Response: We have inserted a paragraph at lines 201-209 that describe some of our operational definitions and sources of disease related concepts, what we mean by drivers of disease and climatic and non-climatic drivers of disease systems.

6. Text in lines 185-187 and the PRISMA flowchart are better suited to the Results section.

Response: We have moved the text in lines 212-215 and the PRISMA flowchart to the beginning of the Results section in the revised manuscript (see line 216, page 10 of the revised manuscript). 

Results & Discussion

7. Overall, authors have clearly reported and critically reflected upon results, and I was pleased to see integration of findings from scientific articles and policy documents.

Response: Thank you for the positive assessment of our paper. 

8. I believe that in Section 3.3 there is again confusion regarding terminology, as some of the so-called hazards (eg., extreme weather events) are referred to as climate drivers. I am aware that there might be vocabulary-related discrepancies between the climate science community and the disaster science community. Perhaps, for the sake of clarity, it would be good to cite sources of definitions. Reporting operational definitions in the methods, as suggested above, might help.

Response: We have inserted a paragraph at lines 201-209 that describes some of our operational definitions and sources of disease related concepts, what we mean by drivers of disease and climatic and non-climatic drivers of disease systems.

9. I was wondering whether “national/state level health surveillance or information systems” (lines 476-477) is limited to systems and mechanisms implemented by the state, or whether you adopted a broader definition of health system that also included non-state actors such as international humanitarian organization, possibly relevant for countries like Afghanistan. Perhaps a reference to this issue might be of interest in section 3.4?

Response: Thanks for the comment. To clarify, we adopted a broader definition of health system to include surveillance or information systems operationalised by non-state actors to ensure we do not inadvertently miss any information/surveillance systems. Examples of such surveillance/information systems captured thanks to our broader approach are the Global Flood Awareness system (GLOFAS, Global future climate info system (provided by the World Bank) and Disease Early Warning System (DEWS) and Response (Afghanistan) operationalised by multilateral non-state actors. 

10. I believe only considering articles - esp. local policies - written in English is a very important limitation. If it is not possible to include documentation written in the local languages, I would advise specifying where possible the number of records that were excluded for language reasons and in any case to reflect on possible selection bias, and implications, in the results/discussion.

Response: Thanks for the comment. We appreciate that the exclusive focus on policies written in English might have resulted in missing some relevant policy documents (e.g. regional/state policies) and peer-reviewed papers published I local languages. Indeed, given that English is one of the official languages for reporting government communications (including policies) and health research in the focal countries, we filtered our Web of Science, PubMed and Scopus literature searches to only display results of papers published in English only. We already alluded to the potential selection bias in the under the limitations section of the original manuscript. Nonetheless, in response to the reviewer’s comment, we have added additional text further clarify this limitation of the study (see lines 586-588, page 34 of the revised manuscript). 

Thank you for the feedback on our manuscript.

---

## [Decision Letter · Decision Letter 1]

13 Aug 2024

Informing climate-health adaptation options through mapping the needs and potential for integrated climate-driven early warning forecasting systems in South Asia – a scoping review

PONE-D-24-09817R1

Dear Dr. Asaaga,

We’re pleased to inform you that your manuscript has been judged scientifically suitable for publication and will be formally accepted for publication once it meets all outstanding technical requirements.

Kind regards,

Md Nazirul Islam Sarker, PhD

Academic Editor

PLOS ONE

Additional Editor Comments (optional):

The author is advised to stay in contact with the production team for the next steps in the publication process.

Reviewers' comments:

Reviewer's Responses to Questions

**Comments to the Author**

1. If the authors have adequately addressed your comments raised in a previous round of review and you feel that this manuscript is now acceptable for publication, you may indicate that here to bypass the “Comments to the Author” section, enter your conflict of interest statement in the “Confidential to Editor” section, and submit your "Accept" recommendation.

Reviewer #1: All comments have been addressed

Reviewer #2: All comments have been addressed

2. Is the manuscript technically sound, and do the data support the conclusions?

Reviewer #1: Yes

Reviewer #2: Yes

3. Has the statistical analysis been performed appropriately and rigorously? 

Reviewer #1: N/A

Reviewer #2: Yes

4. Have the authors made all data underlying the findings in their manuscript fully available?

Reviewer #1: Yes

Reviewer #2: Yes

5. Is the manuscript presented in an intelligible fashion and written in standard English?

Reviewer #1: Yes

Reviewer #2: Yes

6. Review Comments to the Author

Reviewer #1: Authors have addressed all the points raised in my first round of revision and I believe the article is now ready for publication.

Reviewer #2: (No Response)

7. PLOS authors have the option to publish the peer review history of their article (what does this mean?). If published, this will include your full peer review and any attached files.

Reviewer #1: **Yes: **Martina Valente

Reviewer #2: No

---

## [Editor Report · Acceptance letter]

21 Aug 2024

PONE-D-24-09817R1 

PLOS ONE

Dear Dr. Asaaga, 

I'm pleased to inform you that your manuscript has been deemed suitable for publication in PLOS ONE. Congratulations! Your manuscript is now being handed over to our production team.

Kind regards, 

on behalf of

Dr. Md Nazirul Islam Sarker 

Academic Editor

PLOS ONE